# SLTUnet: A Simple Unified Model for Sign Language Translation

**Biao Zhang**[1]**, Mathias Müller**[2]**, Rico Sennrich**[2,1]

[1] School of Informatics, University of Edinburgh
[2] Department of Computational Linguistics, University of Zurich
`b.zhang@ed.ac.uk,mmueller@cl.uzh.ch,sennrich@cl.uzh.ch`

## Abstract

Despite recent successes with neural models for sign language translation (SLT), translation quality still lags behind spoken languages because of the data scarcity and modality gap between sign video and text. To address both problems, we investigate strategies for cross-modality representation sharing for SLT. We propose SLTUnet, a simple unified neural model designed to support multiple SLT-related tasks jointly, such as sign-to-gloss, gloss-to-text and sign-to-text translation. Jointly modeling different tasks endows SLTUnet with the capability to explore the cross-task relatedness that could help narrow the modality gap. In addition, this allows us to leverage the knowledge from external resources, such as abundant parallel data used for spoken-language machine translation (MT). We show in experiments that SLTUnet achieves competitive and even state-of-the-art performance on PHOENIX-2014T and CSL-Daily when augmented with MT data and equipped with a set of optimization techniques. We further use the DGS Corpus for end-to-end SLT for the first time. It covers broader domains with a significantly larger vocabulary, which is more challenging and which we consider to allow for a more realistic assessment of the current state of SLT than the former two. Still, SLTUnet obtains improved results on the DGS Corpus. Code is available at https://github.com/bzhangGo/sltunet.

## 1 Introduction

The rapid development of neural networks opens the path towards the ambitious goal of universal translation that allows converting information between any languages regardless of data modalities (text, audio or video) (Zhang, 2022). While the translation for spoken languages (in text and speech) has gained wide attention (Aharoni et al., 2019; Inaguma et al., 2019; Jia et al., 2019), the study of sign language translation (SLT) – a task translating from sign language videos to spoken language texts – still lags behind despite its significance in facilitating the communication between Deaf communities and spoken language communities (Camgoz et al., 2018; Yin et al., 2021). SLT represents unique challenges: it demands the capability of video understanding and sequence generation. Unlike spoken language, sign language is expressed using hand gestures, body movements and facial expressions, and the visual signal varies greatly across signers, creating a tough modality gap for its translation into text. The lack of supervised training data further hinders us from developing neural SLT models of high complexity due to the danger of model overfitting.

Addressing these challenges requires us to develop inductive biases (e.g., novel model architectures and training objectives) to enable knowledge transfer and induce universal representations for SLT. In the literature, a promising way is to design unified models that could support and be optimized via multiple tasks with data from different modalities. Such modeling could offer implicit regularization and facilitate the cross-task and cross-modality transfer learning that helps narrow the modality gap and improve model's generalization, such as unified vision-language modeling (Jaegle et al., 2022; Bao et al., 2022; Kaiser et al., 2017), unified speech-text modeling (Zheng et al., 2021; Tang et al., 2022; Bapna et al., 2022), multilingual modeling (Devlin et al., 2019; Zhang et al., 2020; Xue et al., 2021), and general data modeling (Liang et al., 2022; Baevski et al., 2022). In SLT, different annotations could be paired into different tasks, including the sign-to-gloss (Sign2Gloss), the sign-to-text (Sign2Text), the gloss-to-text (Gloss2Text) and the text-to-gloss (Text2Gloss) task. These

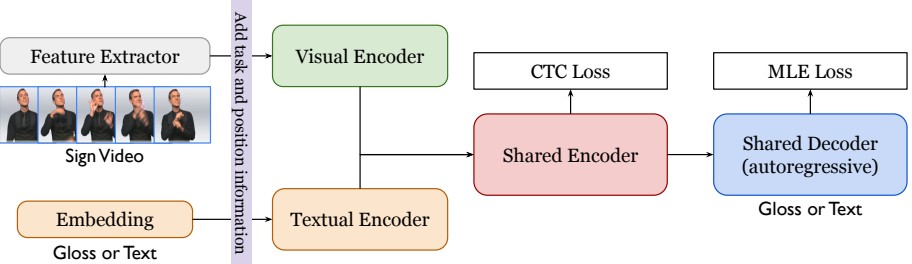

(a) SLTUNET Illustration

| Task | Task Tag | Input | Output | Training Objective |
|---|---|---|---|---|
| Sign2Gloss | *[2gls]* | sign video | gloss | $\alpha\mathcal{L}^{\text{CTC}}(\text{gloss}) + \mathcal{L}^{\text{MLE}}(\text{gloss})$ |
| Sign2Text | *[2txt]* | sign video | text | $\alpha\mathcal{L}^{\text{CTC}}(\text{gloss}) + \mathcal{L}^{\text{MLE}}(\text{text})$ |
| Gloss2Text | *[2txt]* | gloss | text | $\mathcal{L}^{\text{MLE}}(\text{text})$ |
| Text2Gloss | *[2gls]* | text | gloss | $\mathcal{L}^{\text{MLE}}(\text{gloss})$ |
| Machine Translation | *[2txt]* | source text | target text | $\mathcal{L}^{\text{MLE}}(\text{target})$ |

(b) Tasks Explored

Figure 1: Overview of the proposed SLTUNET and the tasks we explored. SLTUNET adopts separate encoders to capture modality-specific (visual and textual) characteristics followed by a shared encoder to induce universal features. It employs an autoregressive decoder shared across tasks for generation. SLTUNET optimizes the whole model via the maximum likelihood estimation (MLE) objective and optionally the connectionist temporal classification (CTC) objective and uses Transformer as its backbone. It supports multiple tasks, such as the sign-to-gloss (Sign2Gloss), the sign-to-text (Sign2Text), the gloss-to-text (Gloss2Text), the text-to-gloss (Text2Gloss) and the machine translation task. We regard the embedding of the corresponding task tag (*[2gls]* or *[2txt]*) as the task information to guide the generation, and append it in front of the input feature sequence inspired by multilingual NMT. $\alpha$ is a hyperparameter; blocks in colour (except gray) indicate trainable parameters; note Text2Gloss hurts SLT in our experiments and is not involved in the final joint objective.

tasks are often modelled separately. Whether unified modeling for them could benefit SLT and what inductive biases are adequate for SLT are still open questions, which are the exact focus of this study.

In this paper, we propose a simple unified model for SLT, namely SLTUNET, to answer the above questions. As in Figure 1, SLTUNET follows the encoder-decoder paradigm (Bahdanau et al., 2015) with Transformer (Vaswani et al., 2017) as its backbone and supports multiple vision/language-to-language generation tasks. It uses shared modules to encourage knowledge transfer and adopts separate visual/textual modules to avoid task or modality interference (Liang et al., 2022). Thanks to its unified schema, SLTUNET allows us to leverage external data resources from other related tasks, such as machine translation. This partially alleviates the data scarcity issue and opens up the possibility of exploring relatively larger models for SLT. We further examine and develop a set of optimization techniques to ensure the trainability of SLTUNET.

We conducted extensive experiments on two popular benchmarks, PHOENIX-2014T (Camgoz et al., 2018) and CSL-Daily (Zhou et al., 2021) for German and Chinese Sign Language, respectively. Following previous evaluation protocols (Camgoz et al., 2018), we test SLTUNET on several SLT-related tasks but *with a single trained model*. Results show that SLTUNET achieves competitive and even state-of-the-art performance, surpassing strong baselines adopting pretrained language models.

We note that PHOENIX-2014T and CSL-Daily, while offering a valuable testbed for SLT, are limited in various aspects. They feature a small number of signers, and are limited in linguistic variety with a small vocabulary. As a more challenging, larger-scale SLT dataset, we propose to use the Public DGS Corpus (Hanke et al., 2020a) that covers broader domains and more open vocabularies, and gives a more realistic view of the current capability of SLT. We also take care in following best practices regarding preprocessing and evaluation (Müller et al., 2022). We find that the challenging nature of the DGS Corpus results in generally low SLT performance, but we still observe some quality gains with SLTUNET. Our contributions are summarized below:

- We propose a simple unified model, SLTUNET, for SLT, and show that jointly modeling multiple SLT-related tasks benefits the translation.
- We propose a set of optimization techniques for SLTUNET aiming at an improved trade-off between model capacity and regularization, which also helps SLT models for single tasks.
- We use the DGS Corpus and propose a translation protocol for end-to-end SLT, with larger scale, richer topics and more significant challenges than existing datasets.
- SLTUNET performs competitively to previous methods and yields the new state-of-the-art performance on CSL-Daily.

## 2  RELATED WORK

Our study on SLT focuses on transforming a sign language video to a spoken language text. Previous methods can be roughly classified into two categories: *cascading* and *end-to-end*.

The cascading method relies on an intermediate output such as sign glosses (Camgoz et al., 2018) where each gloss is a manual transcription for a sign to reflect its meaning. Cascading systems break SLT down into two separate tasks: *sign language recognition* that transcribes a continuous sign video to a gloss sequence (Sign2Gloss) and *gloss-to-text translation* that transforms the glosses to a spoken language text (Gloss2Text). Sign2Gloss requires the modeling of spatial-temporal relations of a sign video to achieve video understanding, which often demands advanced optimizations and architectures, such as 2D/3D-convolutional or recurrent encoders (Cui et al., 2017; Koller et al., 2020), spatial-temporal multi-cue network (Zhou et al., 2022), self-mutual distillation learning (Hao et al., 2021), and cross-modality augmentation (Pu et al., 2020), etc. By contrast, Gloss2Text resembles machine translation (MT) but suffers greatly from data scarcity (Yin & Read, 2020). Recent studies often explore techniques from MT to alleviate this problem, such as data augmentation (Moryossef et al., 2021; Angelova et al., 2022) and using pretrained language models (De Coster et al., 2021; Cao et al., 2022). Unfortunately, sign glosses are not equivalent to their corresponding sign video and often drop information. This imposes a hard performance cap on cascading SLT.

We thus focus on the end-to-end method instead, which converts sign videos directly to natural texts (Sign2Text). Camgoz et al. (2018) pioneered this direction by framing the task as a neural MT problem and showed the feasibility of the encoder-decoder paradigm (Bahdanau et al., 2015). Later studies followed this paradigm and put efforts into improving the sample efficiency and reducing the vision-language modality gap. Camgoz et al. (2020a) and Zhou et al. (2022) developed multi-channel neural models to leverage information from different visual cues (such as hand shapes and facial expressions) to enhance sign language understanding. Li et al. (2020) and Kan et al. (2022) proposed hierarchical neural models to capture spatio-temporal features at multiple levels of granularity in sign videos. Zhou et al. (2021) explored sign back-translation to construct pseudo-parallel training data for SLT based on monolingual texts. Jin et al. (2022) investigated the use of external prior knowledge. Different from the above studies, we focus on unified modeling for SLT with the goal of transferring knowledge across different tasks and particularly improving Sign2Text.

Our study is closely related to multi-modality transfer learning (Chen et al., 2022), with significant differences. Chen et al. (2022) employ Sign2Gloss and Gloss2Text tasks to perform in-domain pretraining for public large-scale pretrained visual and language models, respectively, followed by a specific finetuning on Sign2Text. Their method follows the pretraining-finetuning paradigm and focuses on adapting pretrained models to SLT instead of joint unified modeling and multitask learning. Note, we train SLTUNET on multiple tasks without relying on pretrained language models and SLTUNET achieves state-of-the-art results on CSL-Daily. Although using sign glosses to regularize the neural encoder is popular in Sign2Text (Camgoz et al., 2020b; Zhou et al., 2021; Chen et al., 2022), the study of jointly modeling multiple SLT-related tasks (>2 tasks) via a single network and the exploration of MT data to improve Sign2Text have never been investigated before.

## 3  SLTUNET MODEL

We aim to design a unified model for SLT to improve the translation by utilizing diverse SLT-related tasks. To this end, we propose SLTUNET which supports general vision/language-to-language generation tasks. Figure 1a illustrates the overall architecture and Figure 1b summarizes the tasks we

explored. Note the design of SLTUNET considers the capacity trade-off practice (Zhang et al., 2021; Liang et al., 2022) with the majority of parameters shared for knowledge transfer while the rest kept separate to capture modality-specific features.

SLTUNET follows the encoder-decoder framework and models the conditional generation probability. In general, it takes as input a task tag $tag$ informing the model which task it's handling and a feature sequence $\mathbf{X} \in \mathbb{R}^{|X| \times d}$ and then builds a neural network to predict the ground-truth reference sequence $Y = \{y_1, y_2, \cdots, y_{|Y|}\}$,

$$\mathbf{X}^O = \text{Encoder}^S \circ \text{Encoder}^P (\mathbf{X}, tag), \quad \mathbf{Y}^O = \text{Decoder}(\mathbf{Y}^I, \mathbf{X}^O), \quad (1)$$

where $|\cdot|$ and $d$ denote the sequence length and model dimension respectively, and $\mathbf{Y}^I \in \mathbb{R}^{|Y| \times d}$ is the right-shifted input feature sequence used for autoregressive decoding. $\circ$ represents the chaining of two modules. $\mathbf{X}^O \in \mathbb{R}^{|X| \times d}$ and $\mathbf{Y}^O \in \mathbb{R}^{|Y| \times d}$ are the encoder and decoder outputs, respectively.

We adopt Transformer (Vaswani et al., 2017) as the backbone for SLTUNET. Decoder$(\cdot)$ indicates the autoregressive Transformer decoder with $N_{dec}$ layers; Encoder$^S(\cdot)$ and Encoder$^P(\cdot)$ stand for shared and modality-specific Transformer encoders with $N_{enc}^S$ and $N_{enc}^P$ layers, respectively. Inspired by multilingual neural MT (Johnson et al., 2017), we append the embedding of $tag$ in front of $\mathbf{X}$ along the time axis and feed the concatenated sequence to the encoder.

Different tasks have different training objectives and different ways to construct $\mathbf{X}$. Depending on the input modality to the encoder, SLTUNET have the following two working modes:

1) **When the task has no sign video inputs,** Encoder$^P(\cdot)$ denotes the textual encoder in Figure 1a and the input feature $\mathbf{X}$ is obtained via a word embedding layer. We train SLTUNET via the following objective:

$$\mathcal{L}(Y|X, tag) = \mathcal{L}^{\text{MLE}}(Y|\mathbf{Y}^O), \quad (2)$$

where $X$ denotes the input of $\mathbf{X}$, $\mathcal{L}^{\text{MLE}}(\cdot)$ is the maximum likelihood estimation (MLE) objective.

2) **Otherwise,** Encoder$^P(\cdot)$ denotes the visual encoder in Figure 1a and we prepare sign embeddings $\mathbf{X}$ based on some pretrained visual models. In particular, we adopt the SMKD model (Hao et al., 2021) and extract its visual features, i.e. the output of 1D temporal convolution, as the sign features. We further project these features to the model dimension via a linear layer to form $\mathbf{X}$. Note the parameters of SMKD are frozen when training SLTUNET. The training objective is:

$$\mathcal{L}(Y, Z|X, tag) = \mathcal{L}^{\text{MLE}}(Y|\mathbf{Y}^O) + \alpha \mathcal{L}^{\text{CTC}}(Z|\mathbf{X}^O), \quad (3)$$

where $\mathcal{L}^{\text{CTC}}(\cdot)$ is the connectionist temporal classification (CTC) objective (Graves et al., 2006) and $Z$ denotes the gold label sequence for CTC, which is often the gloss sequence in SLT. Different from MLE, CTC models the probability distribution by marginalizing over all valid mappings between its input ($\mathbf{X}^O$) and output ($Z$) sequence. CTC has been widely used in SLT to regularize the sign encoder (Camgoz et al., 2018; Chen et al., 2022), and we follow this practice and use a hyperparameter $\alpha$ to balance its effect. Note the CTC part will be dropped after training.

As shown in Figure 1b, SLTUNET offers high flexibility to accommodate different SLT-related tasks. Also, it allows us to explore knowledge from other tasks by leveraging their abundant training data, such as machine translation. Formally, given a SLT training sample (sign video, gloss sequence, text translation) denoted by $(\mathcal{V}, \mathcal{G}, \mathcal{T})$ and a MT sample (source text, target text) denoted by $(S, T)$, the final SLTUNET training objective is formulated below:

$$\mathcal{L}^{\text{SLTUNET}} = \underbrace{\mathcal{L}(\mathcal{G}, \mathcal{G}|\mathcal{V}, [2gls])}_{\text{Sign2Gloss}} + \underbrace{\mathcal{L}(\mathcal{T}, \mathcal{G}|\mathcal{V}, [2txt])}_{\text{Sign2Text}} + \underbrace{\mathcal{L}(\mathcal{T}|\mathcal{G}, [2txt])}_{\text{Gloss2Text}} + \underbrace{\mathcal{L}(T|S, [2txt])}_{\text{Machine Translation}}, \quad (4)$$

where we adopt a multi-task learning schema and treat different tasks equally for training. Note, we exclude Text2Gloss in the final objective and only retain the CTC objective for Sign2Text based on our preliminary experiments, and we mix SLT and MT samples during training based on a predefined ratio. At testing, we examine SLTUNET under two modes – the end-to-end (Sign2Text) and cascading (Sign2Gloss + Gloss2Text) mode – *using a single trained model*.

**Optimization for SLTUNET** Covering multiple tasks entails more training data and reduced risk of model overfitting. This gives us the opportunity to increase the modeling capacity for SLTUNET by adjusting the model depth and width. Meanwhile, we still need to control the degree of model regularization via e.g., dropout rates to achieve the full potential of SLTUNET. All these make the optimization of SLTUNET challenging and we will examine different methods in the experiments.

| Dataset | Lang | Attribute | | | Statistics | | | | | |
|---|---|---|---|---|---|---|---|---|---|---|
| | | Resolution | Doc. | #Signers | Vocab | #OOV | #Train | #Dev | #Test | |
| PHOENIX-2014T | DGS | $210 \times 260$ | ✗ | 9 | 1,085/2,887 | 30/113 | 7,096 | 519 | 642 | |
| CSL-Daily | CSL | $1920 \times 1080$ | ✗ | 10 | 2,000/2,277 | 0/37 | 18,401 | 1,077 | 1,176 | |
| DGS3-T | DGS | $640 \times 360$ | ✓ | 330 | 8,580/23,363 | 105/647 | 60,306 | 967 | 1,575 | |

Table 1: Summary of different SLT datasets. *Lang*: language; *DGS*: German Sign Language; *CSL*: Chinese Sign Language; *Doc.*: whether samples are organized in the form of document; *#Signers*: number of individuals in the entire dataset; *Vocab*: number of glosses/spoken words in the training set (note we count characters for Chinese); *#OOV*: out-of-vocabulary glosses/words that occur in dev and test sets but not in the train set; *#Train/#Dev/#Test*: number of samples in the train/dev/test set, respectively.

## 4 EVALUATING END-TO-END SYSTEMS ON LARGER-SCALE DATA

Although popular benchmarks PHOENIX-2014T (Camgoz et al., 2018) and CSL-Daily (Zhou et al., 2021) offer a valuable testbed for SLT, we note that they suffer from limitations such as training data size, the size of their gloss and spoken language vocabulary, the number of signers, and domains and topics covered as shown in Table 1. For example, both benchmarks feature a vocabulary of <3000 spoken language words, representing just a fraction of the vocabulary typical in spoken language MT systems. Thus, existing results may give too rosy an impression of the current capability of SLT models.

We therefore use the Public DGS Corpus (Hanke et al., 2020b), as a broader-domain and more realistic testbed. The DGS Corpus is a dataset featuring German Sign Language (DGS), German and English. It includes data collected from 330 signers from 12 different locations in Germany. The signers were balanced for gender, age, and region, and the data covers various linguistic domains (such as story telling and conversations). Whereas previous work has focused on Gloss2Text (Müller et al., 2022; Angelova et al., 2022), our focus lies in evaluating and improving the Sign2Text task.

We create a document-level dataset split, which offers room to study contextual modeling in the future. The split contains 60,306, 967, and 1,575 samples in the train, dev, and test set, respectively (see Table 1 and Appendix A.2 for details). We will refer to this dataset as DGS3-T for short, referring to the fact that we use release 3 of the Public DGS Corpus and that we use it for translation tasks ("T") rather than vision tasks such as sign language production (Saunders et al., 2022). Similar to previous SLT datasets, each sample in DGS3-T is a triplet consisting of a sign video, sentence-level gloss annotation and the German translation (besides other annotations). DGS3-T has a large vocabulary with 8,580 glosses and 23,363 spoken language words, posing considerable practical challenges.

## 5 EXPERIMENTS

### 5.1 SETUP

**Datasets** We work on three SLT datasets: *PHOENIX-2014T*, *CSL-Daily*, and *DGS3-T*. PHOENIX-2014T and DGS3-T focus on German Sign Language, CSL-Daily on Chinese Sign Language. All three datasets provide triplet samples, each consisting of a sign language video, a sentence-level gloss annotation and their corresponding text translation. Detailed statistics are listed in Table 1. We employ MuST-C English-German (En-De, 229K samples) and English-Chinese (En-Zh, 185K samples) (Di Gangi et al., 2019) as the augmented MT data for PHOENIX-2014T/DGS3-T and CSL-Daily, respectively. We learn a joint vocabulary for glosses and texts via byte pair encoding (BPE) (Sennrich et al., 2016). We employ 1K BPE operations when MT data is not used, and increase it to 8K/8K/10K for PHOENIX-2014T/DGS3-T/CSL-Daily otherwise.

**Model Settings** We experiment with Transformer (Vaswani et al., 2017) and start our analysis with a *Baseline* system optimized on Sign2Text alone with the following configurations: encoder and decoder layers of $N_{enc}^S = 2, N_{enc}^P = 0$ and $N_{dec} = 2$ respectively, model dimension of $d = 512$, feed-forward dimension of $d_{ff} = 2048$, attention head of $h = 8$, and no CTC regularization. We

| ID | System | #params | B@4↑ |
|----|--------|---------|------|
| 1 | Baseline | 15.8M | 22.62 |
| 1.1 | 1 + sign embeddings from (Camgoz et al., 2020b) | 15.8M | 21.21 |
| *Explore CTC Regularization* | | | |
| 2 | 1 + CTC loss ($\alpha = 0.3$) | 16.3M | 24.04 |
| | 2 + relative positional encoding (Shaw et al., 2018) ($k = 16$) | 16.3M | 23.92 |
| | 2 + $\alpha = 0.2$ | 16.3M | 23.71 |
| | 2 + $\alpha = 0.4$ | 16.3M | 23.79 |
| *Explore Multi-task Learning* | | | |
| 3 | 2 + multi-task training (Equation 4 without MT) | 16.3M | 25.10 |
| 3.1 | 3 + add Text2Gloss for training | 16.3M | 24.88 |
| 3.2 | 3 + remove Sign2Gloss at training | 16.3M | 23.96 |
| 3.3 | 3 + remove Gloss2Text at training | 16.3M | 24.60 |
| 4 | 3 + add MT task (mixing ratio for MT and SLT samples 3:1, vocab size: 1K → 8K) | 23.6M | 25.23 |
| *Explore Modality-Specific Modeling* | | | |
| 5 | 4 + add modality-specific module ($N_{enc}^{P} = 1, N_{enc}^{S} = 2, N_{dec} = 3$) | 34.1M | 26.30 |
| 5.1 | 5 + add more modality-specific parameters ($N_{enc}^{P} = 2, N_{dec} = 4$) | 44.6M | 25.41 |
| | 5 + change mix ratio from 3:1 to 5:1 | 34.1M | 25.95 |
| | 5 + apply CTC regularization to the output of visual encoder instead | 34.1M | 25.48 |
| *Explore Model Regularization* | | | |
| 6 | 5 + apply BPE dropout (Provilkov et al., 2020) to glosses and texts of rate 0.2 | 34.2M | 26.04 |
| | 6 + increase BPE dropout rate to 0.3 | 34.2M | 25.67 |
| | 6 + decrease BPE dropout rate to 0.1 | 34.2M | 26.00 |
| 7 | 6 + stochastic BPE dropout of stochastic rate 0.5 | 34.2M | 26.50 |
| 8 | 7 + apply random crop and horizontal flip (50%) to sign video frames for augmentation | 34.2M | 26.76 |
| | 8 + $L2$ weight regularization with a coefficient of $1e^{-3}$ | 34.2M | 26.79 |
| 9 | 8 + change the gain hyperparameter in Xavier initialization to 0.5 | 34.2M | 27.11 |
| *Explore Larger-Capacity Modeling* | | | |
| 10 | 9 + increase model depth ($N_{enc}^{P} = 1, N_{enc}^{S} = 5, N_{dec} = 6$) | 56.2M | 26.56 |
| 11 | 10 + reduce model dimension ($d = 256, h = 4$) | 23.1M | 27.38 |
| | 11 + add more modality-specific parameters ($N_{enc}^{P} = 2, N_{enc}^{S} = 4, N_{dec} = 6$) | 24.5M | 27.13 |
| | 11 + increase feed-forward layer ($d_{ff} = 4096$) | 36.8M | 27.09 |
| 12 | 11 + increase model depth ($N_{enc}^{P} = 1, N_{enc}^{S} = 7, N_{dec} = 8$) | 28.9M | 27.39 |
| | 12 + layer dropout of rate 0.1 | 28.9M | 26.99 |
| *Explore Capacity-Regularization Balance* | | | |
| 13 | 11 + increase stochastic rate to 0.6 | 23.1M | 27.44 |
| 14 | 13 + increase feed-forward layer ($d_{ff} = 4096$) and its dropout rate to 0.5 | 36.8M | 27.56 |
| SLTUNET | | | |
| 15 | 14 + update sign embeddings with improved SMKD model | 36.8M | **27.87** |

Table 2: Ablation study of SLTUNET on Sign2Text on the PHOENIX-2014T dev set. *#params*: number of trainable model parameters; *B@4*: tokenized 4-gram BLEU.

adopt the SMKD model (Hao et al., 2021)[1] to extract sign embeddings, and pretrain the model on each benchmark separately on the Sign2Gloss task considering the large difference of sign videos across benchmarks. More details about datasets and model settings are given in Appendix A.1.

**Evaluation** We report results mainly on the SLT task. Following previous evaluation protocol (Camgoz et al., 2018), we examine our model via the end-to-end (Sign2Text) and cascading (Sign2Gloss +Gloss2Text) method for SLT; we measure the translation performance using tokenized BLEU with n-grams from 1 to 4 (B@1-B@4) (Papineni et al., 2002) and Rouge-L F1 (ROUGE) (Lin, 2004), and we employ Word Error Rate (WER) to evaluate Sign2Gloss.[2]

---

[1] https://github.com/ycmin95/VAC_CSLR
[2] Metric scripts https://github.com/neccam/slt/blob/master/signjoey/metrics.py

We note that the current evaluation practices in SLT do not align with more general MT research, where B@1-B@3 and ROUGE are often considered inadequate to evaluate translation due to their relatively inferior correlation with human judgement. We follow the recent recommendations from MT (Kocmi et al., 2021) and further report detokenized BLEU (sBLEU) and ChrF (Popović, 2015) offered by SacreBLEU (Post, 2018), while we acknowledge that how to properly evaluate translation is still an ongoing research topic. Note we always use character-level metrics for Chinese translation.

## 5.2 RESULTS AND ANALYSIS

We perform our main analyses on PHOENIX-2014T and summarize the results in Table 2.[3]

**High-quality sign embedding and CTC regularization benefit SLT.** Replacing sign embeddings offered by Camgoz et al. (2020b) (24.88 WER↓ on dev set) with the one from our retrained SMKD model (19.80 WER) greatly improves SLT (+1.41 BLEU, 1.1→1). Changing the visual backbone of SMKD to the 2D Resnet34 pretrained on ImageNet (He et al., 2016) delivers further quality gains (18.90 WER, +0.31 BLEU, 14→15). Also, adding CTC regularization helps (+1.42 BLEU, 1→2) resonating with previous findings (Camgoz et al., 2020b). We didn't see obvious benefit from the relative positional representation (Shaw et al., 2018).

**Unified modeling via multi-task learning improves SLT and different tasks show different impacts.** Unified modeling could facilitate knowledge transfer across tasks especially when the tasks are highly correlated. In Table 2, we observe that modeling Sign2Gloss, Sign2Text and Gloss2Text together improves SLT (+1.06 BLEU, 2→3). But adding Text2Gloss deteriorates the performance (-0.22 BLEU, 3→3.1). This might be caused by the large gap between text translation and sign video that hinders the transfer in encoder. Sign2Gloss benefits the unified modeling more than Gloss2Text (3.2 vs. 3.3). Leveraging external resources, such as MT data, also helps SLT though the quality gain is small (+0.13 BLEU, 3→4). We still include MT in SLTUNET since it brings in rich training data that could alleviate overfitting and allow us to explore higher-capacity models.

**Mixing shared parameters with adequate modality-specific parameters improves the transfer.** Sharing parameters across modalities/tasks enables knowledge transfer but often at the cost of cross-modality/task interference partially due to its insufficiency in describing modality/task-specific characteristics (Wang et al., 2019; Liang et al., 2022). Previous studies also showed the trade-off between shared parameters and task-specific parameters in a joint network (Zhang et al., 2021). As shown in Figure 1a, we incorporate modality-specific (visual and textual) encoders to mitigate the interference, which obtains significant quality boost (+1.07 BLEU, 4→5). Further increasing the amount of modality-specific parameters helps little, though (-0.89 BLEU, 5→5.1).

**Unified modeling benefits from an appropriate degree of model regularization.** We next examine a set of regularization techniques for SLTUNET considering the low-resource condition of SLT. BPE dropout regularizes neural models by producing diverse subword segmentations of a word with randomness which greatly improves low-resource MT (Provilkov et al., 2020). Unfortunately, directly applying it to SLTUNET delivers inferior performance (-0.26 BLEU, 5→6). We then propose a simple variant, named **stochastic BPE dropout**, that applies BPE dropout to a random proportion of samples. We empirically set the stochastic rate to 0.5, i.e., only 50% of samples are handled by BPE dropout with the rest retained, which slightly improves SLT (+0.2 BLEU, 5→7).

In image processing, a popular way of regularization is to augment the training data by applying cropping and flipping operations. We follow Hao et al. (2021) and adopt random crop and horizontal flip (50%) to sign frames, which delivers a gain of 0.26 BLEU (7→8). We find that the traditional $L2$ weight decay helps little, but changing the gain parameter in Xavier initialization from 1.0 to 0.5 benefits the translation (+0.35 BLEU, 8→9).

**Larger-capacity modeling via tuning model depth/width with careful regularization further improves translation.** Jointly modeling multiple tasks gives us the chance to explore larger-capacity modeling, but naively increasing model depth (9→10) hurts the performance greatly (-0.55 BLEU). We then reduce the model dimension from 512 to 256 which delivers positive gains (+0.82 BLEU, 10→11). On top of it, we explore increasing modality-specific layers or model depth, reducing

---

[3]Note we explore the near *optimal* setting for SLTUNET mainly based on our experience rather than a full-space grid search. Aggressively optimizing the system might offer better SLT performance but requires massive computing resources that we can't afford.

| Task | Sign2Gloss | Gloss2Text | | Sign2Text | | Cascading | |
|------|------------|------------|------|-----------|------|-----------|------|
| Metric | WER↓ | sBLEU↑ | ChrF↑ | sBLEU↑ | ChrF↑ | sBLEU↑ | ChrF↑ |
| Single Task | **18.36** | 25.42 | 50.52 | 26.56 | 52.03 | 24.48 | 49.49 |
| Single Task w/ MT | 19.54 | 26.69 | 51.61 | 27.36 | 52.72 | 24.54 | 49.83 |
| Multi Task | 19.24 | **27.09** | **52.14** | **27.87** | **53.01** | **25.36** | **50.75** |

Table 3: Ablation results of single-task and multi-task training for SLTUNET with the setup of system 15 in Table 2 on the PHOENIX-2014T dev set. *w/ MT*: augmenting each SLT task with MT; *Cascading*: cascading performance on SLT where we chain a Sign2Gloss model and a Gloss2Text model trained separately for single-task evaluation. Notice that we feed the reference glosses for the Gloss2Text task.

| Task & Systems | Dev | | Test | | | | |
|----------------|-----|-----|------|-----|-----|-----|-----|
| | ROUGE | B@4 | ROUGE | B@1 | B@2 | B@3 | B@4 |
| *Cascading: Sign2Gloss +Gloss2Text* | | | | | | | |
| SL-Transf. (Camgoz et al., 2020b) | - | 22.11 | - | 48.47 | 35.35 | 27.57 | 22.45 |
| BN-TIN-Transf.+BT (Zhou et al., 2021) | 49.53 | 23.51 | 49.35 | 48.55 | 36.13 | 28.47 | 23.51 |
| STMC-Transf. (Yin & Read, 2020) | 46.31 | 22.47 | 46.77 | 48.73 | 36.53 | 29.03 | 24.00 |
| ConSLT (Fu et al., 2022) | - | 24.31 | - | **51.29** | 38.62 | 30.79 | 25.48 |
| VL-Transfer (Chen et al., 2022) | **50.23** | 24.63 | 49.59 | 49.94 | 37.28 | 29.67 | 24.60 |
| SLTUNET (sBLEU: **26.00**, ChrF: **51.96**) | 49.61 | **25.36** | **49.98** | 50.42 | **39.24** | **31.41** | **26.00** |
| *End-to-end: Sign2Text* | | | | | | | |
| SL-Transf. (Camgoz et al., 2020b) | - | 22.38 | - | 46.61 | 33.73 | 26.19 | 21.32 |
| BN-TIN-Transf.+BT (Zhou et al., 2021) | 50.29 | 24.45 | 49.54 | 50.80 | 37.75 | 29.72 | 24.32 |
| STMC-T (Zhou et al., 2022) | 48.24 | 24.09 | 46.65 | 50.80 | 37.75 | 29.72 | 24.32 |
| PET (Jin et al., 2022) | - | - | 49.97 | 49.54 | 37.19 | 29.30 | 24.02 |
| VL-Transfer (Chen et al., 2022) | **53.10** | 27.61 | **52.65** | **53.97** | 41.75 | 33.84 | 28.39 |
| SLTUNET (sBLEU: **28.47**, ChrF: **53.78**) | 52.23 | **27.87** | 52.11 | 52.92 | **41.76** | **33.99** | **28.47** |

Table 4: Results of different systems on PHOENIX-2014T. *B@1-B@4*: tokenized BLEU with n-grams from 1 to 4, respectively. The numbers in bracket for SLTUNET denote sBLEU and ChrF on the test set. Best results are highlighted in **bold**. SLTUNET achieves competitive and even the best performance. Note results from previous papers might not be directly comparable as they might use different tokenizers and evaluation toolkits.

model dimension, and enlarging feed-forward layers, but don't get encouraging results. We argue that adding capacity leads to higher risk of model overfitting thus demanding more regularization. Based on this, we increase the stochastic BPE dropout rate to 0.6 and the feed-forward layer to 4096. This results in an improved system (14) with a BLEU gain of 0.18 (11→14).

**Putting all together, SLTUNET achieves substantial improvements against Baseline.** SLTUNET achieves a BLEU score of 27.87, surpassing Baseline by 5.25 BLEU, a large margin (1→15). Further ablation study in Table 3 shows that the benefits from the unified modeling and multi-task learning are still promising under the optimized setup for SLTUNET. We summarize the final configuration (i.e. system 15 in Table 2) below: $d = 256, h = 4, d_{ff} = 4096, N_{enc}^{P} = 1, N_{enc}^{S} = 5, N_{dec} = 6$, CTC regularization with $\alpha = 0.3$, stochastic BPE dropout with dropout rate of 0.2 and stochastic rate of 0.6, Xavier initialization with gain of 0.5, sign frame augmentation (random crop and horizontal flip), and objective Equation 4. We adopt this setup for next experiments unless otherwise specified.

**SLTUNET achieves (near) the state-of-the-art on two previous benchmarks.** We compare the results of SLTUNET with previous studies on PHOENIX-2014T and CSL-Daily in Table 4 and 5, respectively. Our model produces competitive and even state-of-the-art results on these benchmarks regardless of using the end-to-end or the cascading method. In particular, SLTUNET largely outperforms the previous best system on CSL-Daily by 1.0+ B@4 and 0.8+ ROUGE. We also show sBLEU and ChrF scores on the test set to facilite future research. Note VL-Transfer adopts large-scale pretrained language models and includes much more model parameters than SLTUNET (Chen et al., 2022). This shows the superiority of SLTUNET on sample and parameter efficiency.

| Task & Systems | Dev | | Test | | | | |
|---|---|---|---|---|---|---|---|
| | ROUGE | B@4 | ROUGE | B@1 | B@2 | B@3 | B@4 |
| *Cascading: Sign2Gloss +Gloss2Text* | | | | | | | |
| SL-Transf. (Camgoz et al., 2020b) | 44.18 | 15.94 | 44.81 | 47.09 | 32.49 | 22.61 | 16.24 |
| BN-TIN-Transf.+BT (Zhou et al., 2021) | 48.38 | 19.53 | 48.21 | 50.68 | 36.00 | 26.20 | 19.67 |
| VL-Transfer (Chen et al., 2022) | 51.35 | 21.88 | 51.43 | 50.33 | 37.44 | 28.08 | 21.46 |
| SLTUNET (sBLEU: **23.76**, ChrF: **21.09**) | **52.89** | **22.95** | **53.10** | **54.39** | **40.28** | **30.52** | **23.76** |
| *End-to-end: Sign2Text* | | | | | | | |
| SL-Transf. (Camgoz et al., 2020b) | 37.06 | 11.88 | 36.74 | 37.38 | 24.36 | 16.55 | 11.79 |
| BN-TIN-Transf.+BT (Zhou et al., 2021) | 49.49 | 20.80 | 49.31 | 51.42 | 37.26 | 27.76 | 21.34 |
| VL-Transfer (Chen et al., 2022) | 53.38 | **24.42** | 53.25 | 53.31 | 40.41 | 30.87 | 23.92 |
| SLTUNET (sBLEU: **25.01**, ChrF: **21.99**) | **53.58** | 23.99 | **54.08** | **54.98** | **41.44** | **31.84** | **25.01** |

Table 5: Results of different systems on CSL-Daily. SLTUNET obtains the best test performance.

| Task & Systems | Dev | | Test | | | | | Test | |
|---|---|---|---|---|---|---|---|---|---|
| | ROUGE | B@4 | ROUGE | B@1 | B@2 | B@3 | B@4 | sBLEU | ChrF |
| *Cascading: Sign2Gloss +Gloss2Text* | | | | | | | | | |
| SL-Transformer | 24.38 | 3.00 | 22.13 | **21.30** | **8.69** | 4.13 | 2.21 | 2.21 | **19.33** |
| SLTUNET | **26.40** | **3.49** | **23.24** | 21.00 | 8.65 | **4.25** | **2.29** | **2.28** | 18.96 |
| *End-to-end: Sign2Text* | | | | | | | | | |
| SL-Transformer | 25.37 | 3.13 | 22.50 | 21.53 | 8.32 | 3.85 | 2.00 | 2.00 | 18.55 |
| SLTUNET | **27.95** | **3.94** | **24.53** | **23.11** | **10.05** | **5.13** | **2.81** | **2.82** | **20.56** |

Table 6: Results of different systems on DGS3-T. *SL-Transformer* is a baseline system following SL-Transf. (Camgoz et al., 2020b) with SMKD sign embeddings. SLTUNET still delivers improved translation.

**The DGS Corpus presents unique challenges and SLTUNET still obtains improved performance.** For this experiment, we mix SLT and MT (MuST-C En-De) samples with a ratio of 1:1. Table 6 shows that overall neural SLT models deliver poor results on DGS3-T although SLTUNET still obtains decent quality gains. Based on manual analysis, we find that models suffer greatly from hallucinations where the generation shows limited correlation with the sign video as in Table 9 in the Appendix. The challenge is also reflected in the poor Sign2Gloss result, where SMKD produces a WER↓ score of 67.00 on the dev set. We argue that the large number of signers and the diverse contents present serious challenges in video understanding, and the Zipfian distribution of glosses and words, with most occurring fewer than 10 times in the training data, as shown in Figure 2 in the Appendix, further increases the learning difficulty.

## 6 CONCLUSION AND FUTURE WORK

In this paper, we explore unified modeling for SLT with the objective to transfer knowledge across tasks and particularly to benefit SLT. We present SLTUNET, a simple encoder-decoder model that supports multiple SLT-related tasks including Sign2Gloss, Gloss2Text, Sign2Text and machine translation. SLTUNET adopts shared parameters and modality-specific parameters to achieve its best result under a set of optimization techniques. We show in experiments that SLTUNET achieves (near) state-of-the-art performance on traditional benchmarks.

We also emphasize that using a corpus such as the DGS Corpus for end-to-end SLT is more meaningful, as it includes more signers, more glosses, richer topics and more training data, presenting unique challenges to SLT. Our initial results show that previous progress might over-estimate the success of neural models on SLT. Further research is needed to make SLT practical on broader-domain datasets.

In the future, we are interested in exploring large-scale pretrained models and devising larger and multilingual datasets for SLT. We are also interested in studying the feasibility of designing unified models to support translation between any pair of speech, sign and text.

## DATA LICENSING

The license of the Public DGS Corpus[4] does not allow any computational research except if express permission is given by the University of Hamburg.

## ACKNOWLEDGMENTS

We thank the reviewers for their insightful comments. This project has received funding from the Swiss National Science Foundation (project MUTAMUR; no. 176727) and the EU Horizon 2020 project EASIER (grant agreement number 101016982).

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

## A APPENDIX

### A.1 SETUP

**Datasets** PHOENIX-2014T is the first publicly available SLT dataset for German Sign Language (DGS) collected from weather forecasts of the German TV station PHOENIX; CSL-Daily is a Chinese Sign Language (CSL) dataset recording the daily life of the deaf community, covering multiple topics such as family life, medical care, school life and so on.

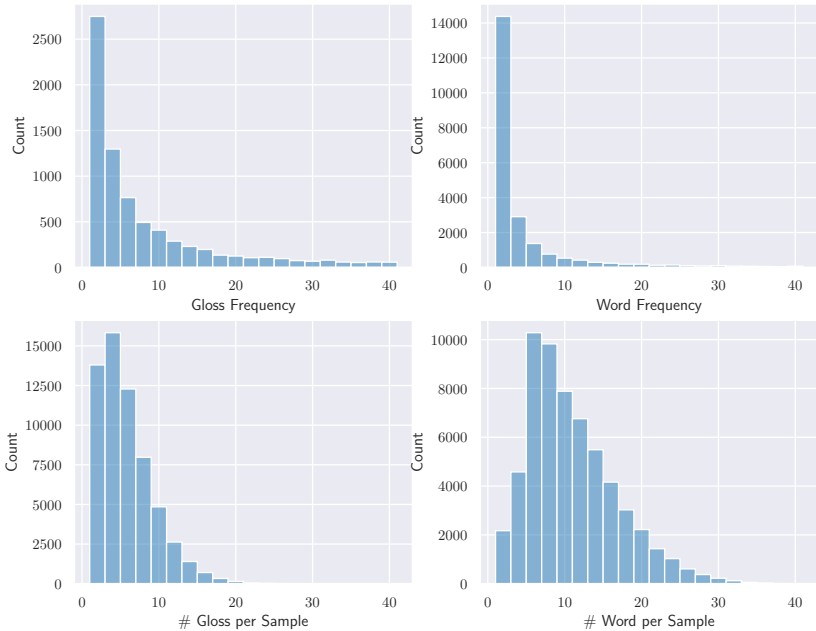

Figure 2: Distribution of gloss frequency, word (in text translation), the number of glosses per sample and the number of words per sample on the DGS3-T train set.

The MuST-C corpus is extracted from TED talks with rich contents from any discipline and culture and has nearly no overlap with the above SLT datasets. This makes it an adequate candidate to study transfer learning for SLT. Note English sentences differ greatly from gloss annotations in grammar, structure and wording. To narrow the gap and facilitate the transfer, we remove punctuation from all English source sentences (Moryossef et al., 2021).

We tokenize all unprocessed texts using Moses (Koehn et al., 2007) and also exclude punctuation from MuST-C German sentences for PHOENIX-2014T.

**Model Settings** We tie the parameters for the input embedding in the textual encoder and the input and softmax embedding in the decoder, and the CTC layer predicts over the shared vocabulary. To avoid overfitting, we apply dropout to the residual connections and feed-forward middle layer of rate 0.4 and to the attention weights of rate 0.3.

We train all SLT models using Adam ($\beta_1 = 0.9, \beta_2 = 0.998$) (Kingma & Ba, 2015) with Noam learning rate schedule (Vaswani et al., 2017), a label smoothing of 0.1 and warmup step of 4K. We employ Xavier initialization to initialize model parameters with a gain of 1.0. We average the best 10 checkpoints based on the dev set result on Sign2Text for the final evaluation. We use beam search for decoding for *all tasks* and set the beam size to 8. We tune the length penalty on the dev set.

**Evaluation** We adopt SacreBLEU (Post, 2018) to report detokenized BLEU (sBLEU) and ChrF. The signatures for sBLEU and ChrF are *BLEU+c.mixed+ #refs.1+s.exp+tok.{13a,zh}+v.1.4.2* and *chrF2+c.mixed+#chars.6+#refs.1+space.False+v.1.4.2*, respectively. Note, on PHOENIX-2014T and CSL-Daily, the value of sBLEU equals to B@4. This is because texts in PHOENIX-2014T are well tokenized with punctuation removed while CSL-Daily uses character-level evaluation. In both cases, tokenization becomes unimportant.

## A.2 DETAILS ON THE DGS3-T TRANSLATION PROTOCOL

We used release version 3 of the Public DGS Corpus (Hanke et al., 2020b). We excluded 2 videos because they have an incorrect framerate of 25 instead of 50. We then randomly assign documents to either the training, development or test split. The desired number of documents in the development and test set is 10. No other preprocessing was performed to create the data split.

| train | validation | | test | |
|:---:|:---:|:---:|:---:|:---:|
| **# signers** | **# signers** | **# unknown** | **# signers** | **# unknown** |
| 328 | 20 | 0 | 20 | 2 |

Table 7: Distribution of signers (individuals) in DGS3-T. # unknown = number of signers that do not appear in the training data.

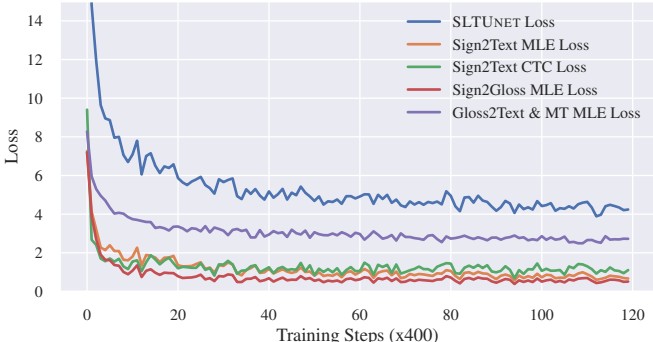

Figure 3: Learning curve of different losses in SLTUNET as a function of training steps on PHOENIX-2014T.

**Identity of signers** The identity of signers has a great impact on models that extract features directly from videos. For tasks involving only glosses and text we assume that the identity of the signer is less important. The overlap of individual signers between training and testing data also matters. We added additional statistics about signers in Table 7. Since most individuals appear in several recording sessions, most signers in our validation and test set are "known". All validation signers also appear in the training set, 18 out of 20 test signers also appear in the training set. Generalization to signers not seen in the training set is known to be more challenging, but the DGS Corpus already has a large number of signers overall (over 300 individuals), improving generalization.

Figure 2 shows distributions of glosses and German words in the data set.

### A.3 ADDITIONAL ANALYSIS

**The convergence of different tasks in SLTUNET follows a similar trend.** Apart from positive knowledge transfer, sharing parameters across tasks might incur inter-task interference hurting the convergence of some tasks. Figure 3 shows the learning curve of different tasks in SLTUNET, which follows a similar trend without obvious convergence disagreement across tasks. This also supports the unified modeling of different SLT-related tasks.

**Aggressive modality-specific modeling hurts SLT performance**. Table 8 shows the results of SLTUNET with either separate encoders or separate decoders. Modeling different modalities with separate modules leads to worse translation results, resonating with our findings in Table 2. Besides, sharing parameters over gloss and text on the decoder side facilitates knowledge transfer, while the transfer between sign video and gloss/text on the encoder side is harder.

### A.4 CASE STUDY FOR PHOENIX-2014T, CSL-DAILY AND DGS3-T

| Task & Systems | Dev | | Test | | | | | Test | |
|---|---|---|---|---|---|---|---|---|---|
| | ROUGE | B@4 | ROUGE | B@1 | B@2 | B@3 | B@4 | sBLEU | ChrF |
| *Cascading: Sign2Gloss +Gloss2Text* | | | | | | | | | |
| SLTUNET | 49.61 | 25.36 | **49.98** | **50.42** | **39.24** | **31.41** | **26.00** | **26.00** | **51.96** |
| + separate encoder | **50.12** | **25.77** | 49.17 | 49.26 | 38.48 | 30.88 | 25.59 | 25.59 | 50.89 |
| + separate decoder | 48.89 | 24.68 | 48.46 | 48.32 | 37.49 | 29.94 | 24.77 | 24.77 | 50.26 |
| *End-to-end: Sign2Text* | | | | | | | | | |
| SLTUNET | **52.23** | **27.87** | **52.11** | **52.92** | **41.76** | **33.99** | **28.47** | **28.47** | **53.78** |
| + separate encoder | 51.62 | 27.64 | 51.73 | 51.95 | 40.86 | 33.08 | 27.62 | 27.62 | 53.50 |
| + separate decoder | 51.52 | 27.19 | 50.95 | 51.03 | 40.38 | 32.76 | 27.33 | 27.33 | 52.75 |

Table 8: Further ablation results of shared and modality-specific modeling for SLTUNET on PHOENIX-2014T. Experiments are based on system 15 in Table 2. *separate encoder*: different encoders for sign video and gloss/text; *separate decoder*: different decoders for gloss and text.

| | |
|---|---|
| Gold Gloss: | MEISTENS1 TAUB-GEHÖRLOS1 BESUCHEN1 WAS1 WÜNSCHEN1 ZIEL4 WAS1 REEPERBAHN1 TYPISCH1 |
| Gold Text: | Die meisten Gehörlosen, die mich besuchen, wollen typischerweise auf die Reeperbahn. (*Most deaf people who visit me typically want to go to the Reeperbahn.*) |
| SLTUNET: | Meistens haben wir Gehörlose besucht und uns wünschen, dass es ein Ziel gibt, eine andere Familie zu bekommen. (*Mostly we visited deaf people and wish that there is a goal to get another family.*) |
| Gold Gloss: | ODER1 $LIST1:2of2 $ALPHA1:S $ALPHA1:M RUND-LANG4 BEKANNT1 $INDEX1 |
| Gold Text: | Die St. Michaelis-Kirche ist auch bekannt für Hamburg. (*St. Michaelis Church is also famous for Hamburg.*) |
| SLTUNET: | Zweitens gibt es den Smartturm, aber das ist nicht berühmt. (*Secondly, there is the smart tower, but that is not famous.*) |
| Gold Gloss: | TURM1 SEHEN1 $PMS TURM1 SEHEN-AUF3 REEPERBAHN1 SEHR-GUT1 BEKANNT1 |
| Gold Text: | Der Fernsehturm und die Reeperbahn, die sind doch bekannt. (*The TV tower and the Reeperbahn are well known.*) |
| SLTUNET: | Wenn die Hörenden in Amerika mehr sind, dann muss man die Hörenden beide beide beide beiden. (*If the hearing in America are more, then you have to have the hearing both both both both.*) |
| Gold Gloss: | MORGEN3 FISCH1 MARKT4 BEKANNT1 $INDEX2 |
| Gold Text: | Morgens geht man zum Fischmarkt, der ist bekannt. (*In the morning you go to the fish market, it's well known.*) |
| SLTUNET: | Ja, das ist bekannt. (*Yes, that is known.*) |

Table 9: Case study for SLTUNET on DGS3-T. Examples are from the test set. The model only translates a tiny part of the input and suffers from hallucinations greatly. Sentences in brackets are our English translations.

| | |
|---|---|
| *Examples from PHOENIX-2014T* | |
| Gold Gloss: | WOCHENENDE IX MEHR KALT |
| Gold Text: | und zum wochenende wird es dann sogar wieder ein bisschen kälter (*and by the weekend it will even be a bit colder again*) |
| SLTUNET: | und am wochenende wird es dann auch wieder kälter (*and on the weekend it will be colder again*) |
| Gold Gloss: | DONNERSTAG NORDWEST REGEN REGION SONNE WOLKE WECHSELHAFT DANN FREITAG AEHNLICH WETTER |
| Gold Text: | am donnerstag regen in der nordhälfte in der südhälfte mal sonne mal wolken ähnliches wetter dann auch am freitag (*on thursday rain in the northern half in the southern half sometimes sunny sometimes cloudy similar weather then also on friday*) |
| SLTUNET: | am donnerstag in küstennähe regen sonst mal sonne mal wolken im wechsel dann am freitag ähnliches wetter (*on thursday rain near the coast otherwise sometimes sun sometimes clouds alternately then similar weather on friday*) |
| Gold Gloss: | SONNTAG NAECHSTE NORDWEST WOLKE SONNE WOLKE GEWITTER REGEN DABEI |
| Gold Text: | am sonntag im nordwesten eine mischung aus sonne und wolken mit einigen zum teil gewittrigen schauern (*on sunday in the northwest a mixture of sun and clouds with some partly thundery showers*) |
| SLTUNET: | am sonntag im norden und westen mal sonne mal wolken mit einzelnen gewittern (*on sunday in the north and west sometimes sunny sometimes cloudy with some thunderstorms*) |
| Gold Gloss: | MORGEN DANN HERBST MISCHUNG HOCH NEBEL WOLKE SONNE |
| Gold Text: | auch morgen erwartet uns eine ruhige herbstmischung aus hochnebel wolken und sonne (*a calm autumn mix of high fog clouds and sun awaits us tomorrow as well*) |
| SLTUNET: | morgen erwartet uns eine meist trübe mischung aus nebel wolken und sonne (*tomorrow we can expect a mostly dull mix of fog clouds and sunshine*) |
| *Examples from CSL-Daily* | |
| Gold Gloss: | 你/ 小/ 张/ 什么/ 时间/ 认识 |
| Gold Text: | 你和小张什么时候认识的？ (*When did you meet Zhang?*) |
| SLTUNET: | 你什么时候认识小张？ (*When did you meet Zhang?*) |
| Gold Gloss: | 今天/ 我/ 想/ 面 |
| Gold Text: | 今天我想吃面条。 (*I want to eat noodles today.*) |
| SLTUNET: | 我今天想吃面条。 (*I want to eat noodles today.*) |
| Gold Gloss: | 今天/ 菜/ 咸/ 我/ 想/ 喝/ 糖 |
| Gold Text: | 今天的菜好咸，我想喝饮料。 (*The food is very salty today, and I want to drink.*) |
| SLTUNET: | 今天的菜很咸，我想喝饮料。 (*The food is very salty today and I want to drink.*) |
| Gold Gloss: | 这/ 男/ 我/ 陌生 |
| Gold Text: | 我不认识那个男生。 (*I don't know that boy.*) |
| SLTUNET: | 那个男生是我的儿子。 (*That boy is my son.*) |

Table 10: Case study for SLTUNET on CSL-Daily and PHOENIX-2014T. Examples are from the test set. Sentences in bracket are our English translations. While SLTUNET achieves better translations on these two benchmarks, it still suffers from difficulties with sign video understanding and delivers inadequate outputs.

