# OpenReview forum: "SLTUNET: A Simple Unified Model for Sign Language Translation"
_ICLR.cc/2023/Conference — ICLR 2023 poster_

### Official Review · Reviewer_LzzH · 2022-10-23

**Confidence:** 4
**Correctness:** 3
**Technical Novelty And Significance:** 3
**Empirical Novelty And Significance:** 3
**Recommendation:** 5

**Clarity, Quality, Novelty And Reproducibility:**

Clarity: The paper is somewhat confusing in the description of the method and the description of the performance metrics.
Quality: Overall quality is average. The article proposes a simple framework with certain innovations. But there is no focus on key issues in the experimental part, and there are some presentation issues.
Novelty: The revised paper has better innovation. The unified framework proposed in this paper for sign language translation learning is relatively novel, and for the first time, machine translation data is added to assist with sign language training.
Reproducibility: The paper has high reproducibility. The method of the paper is concise, and there are detailed experimental details.

**Strength And Weaknesses:**

Strength：
1. The paper combines the related tasks of sign language translation in a clever way. It is only necessary to perform different processing on different modalities in the input layer, and then a unified network can be used for subsequent training. The method is simple and effective and improves performance without significantly increasing model parameters.
2. This paper has done a lot of ablation experiments, and the experimental details shown in Table 2 can enable readers to understand the improvement brought by each method and enhance the reproducibility of the method.

Weakness:
1. There is a little description of the method in this paper. There is only an illustration chart, and no details of the model are shown. It is not stated how the different modal inputs are fused at each iteration of the model.
2. The experimental part of the paper does not highlight its method. The main innovation of this paper is to use a unified framework for related tasks of sign language translation and expect that multi-task training can improve the performance of the model. However, there is only a small amount of content about multi-task training in the experimental part of the paper.
3. The performance metrics in the paper are confusing. The article discusses the common performance metrics of machine translation and uses two different metrics, tokenized Bleu (B@4) and detokenized Bleu (sBLEU). However, the values of these two indicators in the experiment of the paper are exactly the same, which is Unreasonable unless a unified Tokenizer is used, namely tokenize13a. This needs to be explained in more detail in the paper.

**Summary Of The Paper:**

This paper focuses on how to solve four strongly related tasks in sign language translation (SLT) with a unified framework. They propose SLTUNET, a simple unified neural model designed to support multiple SLT-related tasks jointly. By discriminating between inputs from different modalities, different tasks can be trained on the same network and thus narrowing the gap between modalities. In addition, the paper also introduces machine translation tasks on this basis to solve the problem of small scale of sign language translation datasets. The model is tested on two popular SLT datasets, and it can achieve good experimental results without using pre-trained large models, proving its effectiveness.

**Summary Of The Review:**

The paper has good innovation, has done some original work, and the proposed method is concise and has good reproducibility. But the clarity of the paper is not high, especially regarding the index part. This is related to the specific performance and the effectiveness of the method and must be explained in detail. Until then, I was skeptical about the paper. If you can well answer the above questions, I am apt to accept this paper.

---

> ### Author Response · Authors · 2022-11-10
> **Response to Reviewer LzzH**
>
> Thanks for your insightful comments!
>
> **Q: There is a little description of the method in this paper.  [...] It is not stated how the different modal inputs are fused at each iteration of the model.**
>
> **A:** Note different modal inputs are not fused in SLTUNet but are fed separately to the model according to tasks. For example, when optimizing Sign2Text, only sign embeddings are transferred to the SLTUNet encoder while the textual encoder part is not activated at all. Only either the textual or visual encoder is used depending on the type of input. During training, we first extract task-specific inputs and outputs from a running batch and then compute training losses for each task separately, followed by a summation of these losses to get the multi-task objective for SLTUNet.  To better illustrate the optimization details, we uploaded our source code. Please refer to the “train_fn” in “model/transformer.py” for more details.
>
> **Q: The experimental part of the paper does not highlight its method. There is only a small amount of content about multi-task training in the experimental part of the paper.**
>
> **A:** Please notice that the proposed method, SLTUNet, includes not only multi-task objectives but also optimization techniques. We find that the previous optimization recipe for Sign2Text doesn’t fit SLTUNet. With more training data and increased learning difficulty, we need to adjust modeling capacity, develop modality-specific modeling, and improve regularization to get the most from SLTUNet. Table 2 shows our main ablation process, where we show how progressively adding new techniques benefits or hurts the multi-task training of SLTUNet. We also discussed how different SLT tasks (Sign2Text, Sign2Gloss, Gloss2Text) affect the transfer performance. Please also consider our response to reviewer x36u on this point.
>
>
> **Q: The performance metrics in the paper are confusing. This needs to be explained in more detail in the paper.**
>
> **A:** This is a great point! Note SacreBLEU (sBLEU) is not a new metric but a toolkit that controls for the impact of tokenization and ensures fair comparison. In machine translation, researchers used to adopt different tokenization methods to compute BLEU, and using aggressive tokenization (resulting in more tokens) often leads to overestimated quality gains. This inconsistency also makes the comparison of translation results across papers confusing and meaningless. sBLEU solves this problem by offering an internal tokenizer. Using sBLEU to report BLEU results has become a consensus in the MT community.
>
> We noticed that B@4 and sBLEU produce the same values on PHOENIX-2014T and CSL-Daily. This is because texts in PHOENIX-2014T are well tokenized with punctuations removed, while CSL-Daily adopts character-level evaluation. In both settings, the tokenizer becomes unimportant. However, with a view to practical applications, we do not recommend punctuation stripping as it was done for PHOENIX-2014T, thus we recommend using natural target references along with sBLEU.

---

> > ### Author Response · Authors · 2022-11-14
> > **Any further concerns?**
> >
> > Thanks for your insightful comments again! Please feel free to ask any follow-up questions!

---

### Official Review · Reviewer_x36u · 2022-10-23

**Confidence:** 3
**Correctness:** 4
**Technical Novelty And Significance:** 3
**Empirical Novelty And Significance:** 3
**Recommendation:** 6

**Clarity, Quality, Novelty And Reproducibility:**



Eq.2  and Eq3. can be written in a better format. For example, L(X, Y |tag) generally refers to modeling joint probability (X,Y) given tag. According to the equation description, this equation is modeling the conditional probability of Y given X and tag. The authors can re-formulate Eq.2 and Eq.3 to a conditional probability format (e.g, L(Y|X, tag)) for better clarification.

**Strength And Weaknesses:**

Strength:

1. The performance is promising. The final model does not rely on any pre-trained networks but achieves comparable performance with models with pre-trained networks as initialization.

2. The multi-task objective is intuitive. Generally speaking, the size of datasets for sign language-to-text translation is limited compared with text-to-text translation datasets. Borrowing knowledge from other tasks and external task is a practical solution.

3. The authors also conduct a comprehensive ablation study to see the effects of model settings to the final performance.


Weaknesses:

1. Table 2 shows that the multi-task objective improves results from 24 to 25.23, but the settings of other architecture details increase performance from  25.23 to 27.87.  Although experiment highlights are state-of-the-art performance, the contribution of the proposed multi-task objective is limited.






**Summary Of The Paper:**

This paper proposes a unified sign language translation framework that translates sign videos to natural texts. To better learn visual and text representations, the authors propose to use multi-task objectives to borrow knowledge from external datasets, like machine translation datasets. It uses a shared encoder and a shared decoder to handle gloss-to-text translation, text-to-text translation, sign-to-gloss translation, and sign-to-text translation. In addition to multi-task objectives, the authors also conduct a detailed ablation study to see how different experiment settings contribute to the final performance, like BPE dropout. Experiments show that the final architecture achieves comparable results with state-of-the-art baselines with large pre-trained networks as initialization.

**Summary Of The Review:**

The papers show promising results with only sign-to-text data and limited external data. The paper is well-written and easy to follow. The idea is simple and intuitive. In addition, the detailed ablation study shows how the specific settings contribute to the final performance.

---

> ### Author Response · Authors · 2022-11-10
> **Response to Reviewer x36u**
>
> Thanks for your insightful comments!
>
> **Q: Table 2 shows that the multi-task objective improves results from 24 to 25.23, but the settings of other architecture details increase performance from 25.23 to 27.87. Although experiment highlights are state-of-the-art performance, the contribution of the proposed multi-task objective is limited.**
>
> **A:** This criticism can be levied against a sizeable proportion of research: the effectiveness of technically novel components may be smaller than the effectiveness of careful hyperparameter tuning and enhancing the baseline with techniques from previous work, such as our use of model regularization. However, we do not believe this diminishes our contribution. Rather, we hope that our discussion of the impact of hyperparameters and a broad range of techniques can be seen in a positive light:
> * We believe it provides valuable insights and guidance to future researchers/practitioners in the field of sign language translation. We establish a strong base system that even with only Sign2Text training and pre-trained visual embeddings achieves a B@4 of 26.91 on the PHOENIX-2014T test set, outperforming a number of recent studies on transfer learning.
> * It is further validation of our multi-task objectives and unified architecture: they are effective both before and after applying various other optimizations to the model (compare table 2 and table 3).
>
> **Q: Eq. 2 and Eq. 3 can be written in a better format.**
>
> **A:** We updated the equations following your suggestions.

---

> > ### Author Response · Authors · 2022-11-14
> > **Any further concerns?**
> >
> > Thanks for your insightful comments again! Please feel free to ask any follow-up questions!

---

### Official Review · Reviewer_FxDU · 2022-10-25

**Confidence:** 4
**Correctness:** 3
**Technical Novelty And Significance:** 3
**Empirical Novelty And Significance:** 2
**Recommendation:** 5

**Clarity, Quality, Novelty And Reproducibility:**

The paper is basically clear, except for the set of optimization technologies. Overall, this current submission just focuses on demonstrating performance on standard datasets and lacks in-depth experimental analysis of how multiple tasks interact with each other.


**Strength And Weaknesses:**

Strengths:
The proposed approach allows multiple tasks (e.g., such as sign-to-gloss, gloss-to-text, and sign-to-text translation) related to sign language translation to be simulated jointly for enhancing the performance of sign language translation.

Weaknesses:

1. The existing experiments were insufficient to verify how multiple tasks interact to improve the performance of sign language translation.

2. In End-to-End: Sign2Text in Table 4, the proposed approach just was comparable to the baseline VL-Transfer model in terms of ROUGE and B@4 on the Test set of PHOENIX-2014T.

3. One of the claimed contributions is that the set of optimization technologies for SLTUNET was proposed to improve the trade-off between model capacity and regularization. It was confusing.

Questions:

1. When there is an imbalance between video and gloss input, what is the result?

2. Author claimed that the DGS Corpus includes larger
scale, richer topics, and more significant challenges than existing datasets. How to evaluate them?

3. When the proposed approach was applied to multiple tasks, how about the learning curving of CTC and MLE losses?

4. If there replaced the shared encoder or decoder with a separate encoder or decoder, what about the performance?

5. In Figure 1 (1), What do the different colors of these modules denote?

**Summary Of The Paper:**

This article proposed a unified model for sign language translation, in which multiple tasks are simulated jointly for enhancing the performance of sign language translation.

**Summary Of The Review:**

Please see review in the previous sections.

---

> ### Author Response · Authors · 2022-11-10
> **Response to Reviewer FxDU**
>
> Thanks for your insightful comments!
>
> **Q: When there is an imbalance between video and gloss input, what is the result?**
>
> **A:** In sign language translation, a sign video is often annotated with a sequence of glosses. We consider an imbalance scenario where gloss is unavailable at training and only (sign video, text) pair is given. Under this setup, we retrain SLTUNet on PHOENIX-2014T (including retraining sign embeddings without glosses) and replace gloss with its text counterpart, which achieves a test B@4 and ROUGE score of 25.96 and 49.29 on end-to-end Sign2Text. While suffering from 2.50 B@4 and 2.82 ROUGE reduction, we consider this result still promising which outperforms several baseline systems adopting glosses.
>
> **Q: The DGS Corpus includes larger scale, richer topics, and more significant challenges than existing datasets. How to evaluate them?**
>
> **A:** The topics in the DGS Corpus are richer than the Phoenix data in the sense that Phoenix contains weather reports only, hence a very narrow linguistic domain.  A small set of examples of the topics covered in the DGS Corpus (examples are taken from here: https://www.sign-lang.uni-hamburg.de/meinedgs/ling/start-name_en.html): Family and relatives, work and profession, food, personal hygiene and health, vacation, politics, sports and games.
>
> The increased diversity and scale is also visible in statistics reported in Table 1, showing substantially larger vocabularies and data sizes for DGS3-T. Table 7 in the Appendix shows that DGS3-T contains a large number of signers (330, compared to 10 individuals in the other datasets).
>
> **Q: When the proposed approach was applied to multiple tasks, how about the learning curving of CTC and MLE losses?**
>
> **A:** We added the learning curve of CTC and MLE losses in Figure 3, Appendix. The figure shows that the convergence of different tasks follows a similar trend without obvious inter-task disagreement, which supports our unified modeling for SLT.
>
> **Q: If there replaced the shared encoder or decoder with a separate encoder or decoder, what about the performance?**
>
> **A:** In Table 2, we present the results of increasing the amount of modality-specific parameters on the encoder side, which yields worse performance: dev B@4 decreases from 26.30 to 25.41 (5 -> 5.1). We conducted further experiments to explore separate encoders and decoders. Results on the PHOENIX-2014T test set are as below:
> |Model       |                         sBLEU   |  ChrF |
> |--------------|---------------------------------|---------|
> |SLTUNet                   |        28.47  |     53.78 |
> |+ separate encoder  |        27.62  |     53.50 |
> |+ separate decoder  |        27.33  |     52.75 |
>
> Overall, SLTUNet performs the best. More results are available in Table 7, Appendix.
>
> **Q: In Figure 1 (1), What do the different colors of these modules denote?**
>
> **A:** Colored modules except gray denote that it includes trainable model parameters. Different colors refer to different modules: red for the shared encoder, blue for the shared decoder, green for the visual encoder, yellow for the textual encoder (including word embedding), and purple for the task tag embedding.
>
> **Regarding the question of “how multiple tasks interact with each other”**, we hope our new results on learning curves and separate encoder/decoder could partially address it. We see evidence that parameter sharing is not just benefiting the decoder, but that the unified visual/textual encoder also shows knowledge transfer between tasks.
>
> **Regarding the empirical novelty of our study**, based on our knowledge, our study stands out as the first one exploring whether and how unified modeling benefits SLT tasks, and our results demonstrate its feasibility and promising performance. The proposed multi-task training paradigm and the developed optimization techniques have been rarely investigated before. We hope our findings can facilitate the development of SLT in the future.
>
> *Please reconsider our paper based on the new results.*

---

> > ### Author Response · Authors · 2022-11-14
> > **Any further concerns?**
> >
> > Thanks for your insightful comments again! Please feel free to ask any follow-up questions!

---

### Official Review · Reviewer_JikT · 2022-10-26

**Confidence:** 4
**Correctness:** 3
**Technical Novelty And Significance:** 3
**Empirical Novelty And Significance:** 3
**Recommendation:** 6

**Clarity, Quality, Novelty And Reproducibility:**

- The paper is generally clear. Especially, the ablation studies are quite clear due to the numbering convention.
- Novelty of the paper relies on the exhaustive study of SLT with multiple loss functions, model depth/width, and regularizations. From the point of view of experiments, this paper does provide insights into different components of an SLT model
- In the current state, it looks mostly reproducible. That being said, the inclusion of a codebase would be useful.

**Strength And Weaknesses:**

Strengths
- The proposed model is clearly defined and the ablation studies are carefully structured in Table 2. The detailed discussion of these results is also appreciated.

Weaknesses
- Design the protocol for Public DGS for an end-to-end SLT setup: It seems that the protocol involved splitting the dataset into train, test, and dev sets. Was there more to that process? Furthermore, it would also be useful to include the details of how these splits were conducted. Do speakers overlap among the splits? If yes, does the speaker overlap matter? How does the speaker's identity affect sign language translation? What is the distribution of text tokens and gloss vocabulary across the splits?
- The back translation losses are not useful for SLT (i.e. impact of Text2Gloss in Section 5.2.2), but Zhou et. al. 2021 showed that it was quite useful. That seems contradictory and some more discussion on this would be useful for the readers.
- The choice of visual backbone could impact the final performance. Why was the SMKD model chosen? Chen et. al. 2022 also uses a visual backbone model which is S3D (i.e. different from the one used in this paper). Chen et. al. 2022 also use CTC losses for training the model. As their model have some level of similarity to the proposed model, it would be useful to explicitly talk about the differences.
    - Especially, as Chen et. al. 2022 is also similar to the proposed model in terms of performance, it would have been insightful to see its performance on the DGS3-T baseline in Table 6.

**Summary Of The Paper:**

This paper tackles a challenging problem of translating sign language to natural language. The problem's challenging nature arises from the data scarcity and the modality gap between video and text. The authors approach the problem from the perspective of learning a joint latent space for video and text, guided by multiple loss functions including CTC and MLE for both gloss and text. These approaches are verified by a thorough examination of ablation studies of model capacity and regularization over challenging datasets PHOENIX-2014T and CSL-Daily. The authors also propose an end-to-end translation protocol for SLT by using an even more challenging dataset of the Public DGS corpus that covers broader domains and more open vocabularies.

**Summary Of The Review:**

The detailed study of the model design choices are appreciated and could be useful for the SLT community. Although, some empirical comparisons on the new protocol of the DGS3-T dataset are missing which are crucial for understanding the complete utility of the proposed model.

---

> ### Author Response · Authors · 2022-11-10
> **Response to Reviewer JikT**
>
> Thanks for your insightful comments!
>
> **Q: Regarding questions on the DGS3-T protocol**
>
> **A:** Some details regarding the protocol creation process: we used the release 3 of the corpus.  We excluded 2 videos because they have an incorrect framerate of 25 instead of 50. We then randomly assign documents to either the training, development or test split. The desired number of documents in the development and test set is 10. No other preprocessing was performed to create the data split. Note that we believe this design decision increases the realism and challenge of this dataset compared to previous work that has simplified the (textual) target side with punctuation removal and/or lowercasing. Our code to generate the split will be made publicly available.
>
> The identity of signers has a great impact on models that extract features directly from videos. For tasks involving only glosses and text we assume that the identity of the signer is less important. The overlap of individual signers between training and testing data also matters. We added additional statistics about signers in Table 7 in the Appendix. Since most individuals appear in several recording sessions, most signers in our validation and test set are “known”. All validation signers also appear in the training set, 18 out of 20 test signers also appear in the training set. Generalization to signers not seen in the training set is known to be more challenging, but the DGS Corpus already has a large number of signers overall (over 300 individuals), improving generalization.
>
> Concerning the distribution of tokens and glosses, and their overlaps between training and testing data: such statistics are listed in Table 1 and Figure 2 in the Appendix.
>
> **Q: Regarding the conflict with Zhou et al. (2021) on back-translation**
>
> **A:** Text2Gloss is quite different from back-translation. Zhou et al. (2021) explored sign back-translation which converts a spoken language text back into a sign language video. They found that training sign language translation models on pseudo Sign2Text pairs, i.e. (generated sign video, text), greatly improves performance. In contrast, we found that adding Text2Gloss to a joint system helps little. We see no conflict between these findings.
>
> **Q: Why was the SMKD model chosen? Chen et. al. 2022 also uses a visual backbone model which is S3D (i.e. different from the one used in this paper).**
>
> **A:** Our choice is based on Hao et al. (2021). They report that SMKD produces a WER of 22.4 on the PHOENIX-2014T test set, which is comparable to Chen et al. (2022)’s model using S3D (22.4 vs. 22.45, lower is better). An exhaustive comparison of visual backbone models for SLT is outside the scope of this paper.
>
> **Q: As [Chen et al. (2022)'s] model have some level of similarity to the proposed model, it would be useful to explicitly talk about the differences.**
>
> **A:** We discuss this in the last paragraph of section 2. Let us elaborate here. The objective of Chen et al., (2022) is to get the most out of existing large-scale pretrained models. To achieve this, they used Sign2Gloss and Gloss2Text tasks to progressively adapt the pretrained models, followed by a specific finetuning on Sign2Text. Their framework is pretraining-finetuning and their final model is still a Sign2Text model. In contrast, SLTUNet provides unified modeling that supports all tasks in a single network. Before our study, whether unified modeling is feasible and whether we can obtain better results (than individual modeling) was still unclear. Further, different from Chen et al., (2022), SLTUNet doesn’t rely on any pretrained language models but still obtains promising performance.
>
> **Q: The inclusion of a codebase would be useful.**
>
> **A:** To facilitate your understanding of SLTUNet, we uploaded our source code as supplementary material. We will include more details and instructions, and release our source code upon acceptance.

---

> > ### Comment · Reviewer_JikT · 2022-11-12
> > **Comparison of baselines on the DGS3-T**
> >
> > I appreciate the responses to my questions and the relevant changes in the paper.
> >
> > I still have one concern which was not satisfactorily answered:
> > > Especially, as Chen et. al. 2022 is also similar to the proposed model in terms of performance, it would have been insightful to see its performance on the DGS3-T baseline in Table 6.
> >
> > While it is understandable that comparing with the many possible visual backbones is out of the scope of the paper,
> > > An exhaustive comparison of visual backbone models for SLT is outside the scope of this paper.
> >
> > Chen et. al. 2022 is a very competitive baseline for the other datasets in the paper. It was surprising not to find it in Table 6, especially when the authors talk about the challenging nature of the DGGS3-T dataset.

---

> > > ### Author Response · Authors · 2022-11-14
> > > **Response to the comparison of baselines on DGS3-T**
> > >
> > > We are interested in seeing the results of other approaches on DGS3-T. Unfortunately, the work of Chen et al. (2022) is non-trivial to reproduce due to the lack of source code and underspecified hyperparameters in a complex progressive pretraining pipeline. More generally, we don't consider the work of Chen et al. (2022) to be a mutually exclusive alternative to ours - some of their strategies, such as using large-scale pre-trained language models as initialization, could also be used to extend SLTUNet."
> > >
> > > Also, please notice that our motivation for introducing DGS3-T is to present a more realistic testbed covering broader domains with richer lexicons for SLT. In contrast to the promising results on PHOENIX-2014T and CSL-Daily, SLTUNet delivers poor performance on DGS3-T, suggesting that the success of neural SLT models on narrower-domain datasets might not be transferable to realistic setups. To further illustrate the challenge presented in DGS3-T, we finetuned mBart on its Gloss2Text part **alone** (a text-to-text translation task) which gives a test B@4 score of 5.05 while a simple SL-Transformer model yields 4.08.

---

> > > > ### Author Response · Authors · 2022-11-21
> > > > **Any further concerns?**
> > > >
> > > > Thanks for your insightful comments again! Please feel free to ask any follow-up questions!

---

> > > > ### Comment · Reviewer_JikT · 2022-12-01
> > > > **Response to the Authors**
> > > >
> > > > Thank you for the response. I understand the frustration of not being able to reproduce prior work due to lack of source code etc.
> > > >
> > > > Based on the clarifications and additional text/experiments in response to the other reviewers, I maintain my positive score for this paper.

---

> > > > > ### Author Response · Authors · 2022-12-02
> > > > > **Thanks for your positive feedback**
> > > > >
> > > > > We sincerely thank you again for your efforts and constructive comments that helped improve our paper and also for your kind understanding! We believe that unified modeling as SLTUnet does will be a promising direction for sign language translation and we will surely release our source code to facilitate the research for SLT.

---

### Decision · Program_Chairs · 2023-01-20

**Decision:**

Accept: poster

**Justification For Why Not Higher Score:**

* better experimental rigor would have justified higher score.

**Justification For Why Not Lower Score:**

* interseting and novel topic.

**Metareview: Summary, Strengths And Weaknesses:**

This paper is about translating sign language to natural language.

Strengths:
* very clearly written
* multiple tasks can be jointly trained
* promising performance
Weaknesses:
* Design the protocol for Public DGS
* Experiments and explanations are somewhat lacking

**Note From Pc:**

if the above contains the word "oral" or "spotlight" please see: "oral" presentation means -> notable-top-5% and "spotlight" means -> notable-top-25%. As stated in our emails, we are disassociating presentation type from AC recommendations